# Revisiting Distribution Reconstruction via Sample Adaptability: Diffusion-Guided Data-Free Quantization

## Abstract

Data-Free Quantization (DFQ), which seeks to align a low-bit quantized network $Q$ with its full-precision counterpart $P$ without access to original training data, has attracted growing attention. The core idea is to synthesize reconstructed samples that approximate the underlying distribution of real data. However, we observe that reconstructed samples in existing arts are adaptive only to Q with *specific* bit widths, rather than *all* bit widths, especially ultra-low bit widths. This raises a key challenge: how to rectify distributional information during sample reconstruction to produce desirable samples that generalize across varied quantization levels. In this paper, we revisit distribution reconstruction via sample adaptability, revealing that 1) the desirable samples enjoy benefits of rectifying distribution reconstruction via adaptability information; beyond that, 2) the forward diffusion process is an optimal noise strategy to rectify reconstructed samples for obtaining desirable samples; and propose a novel **Diff**usion-Guided **D**ata-**F**ree **Q**uantization approach, dubbed **DiffDFQ**. Unlike conventional direct optimization, we rectify distribution reconstruction via noise diffusion in a *progressive* approximation manner. Technically, we decompose DFQ into three stages: *sample synthesis* to obtain reconstructed samples; upon that, *sample diffusion* to progressively infuse the noise via *forward diffusion process*, yielding desirable samples for varied $Q$; and *network calibration* to calibrate $Q$ with progressive selection strategy over a series of diffused samples. Our **DiffDFQ** enjoys the appealing insights: 1) the diffused samples exhibit effective balance between distribution reconstruction and sample adaptability to facilitate varied $Q$, especially ultra-low bit widths, *e.g.,* 2 bits and 3 bits; notably, 2) unlike the generator-reliance arts requiring up to 1.2M synthetic samples, **DiffDFQ** synthesizes merely 5.12K (1K) samples to earn performance gain over ImageNet for classification. Our empirical studies verify the merits of **DiffDFQ** over state-of-the-arts for classification across varied bits to $Q$. *Our code is available in the supplementary material package.*

## 1 Introduction

Due to limited training data, privacy issue and other factors, data-free quantization (DFQ) Liu et al. (2021); Cai et al. (2020); Zhu et al. (2021); Choi et al. (2021); Qian et al. (2023a); Zhong et al. (2022); Qin et al. (2023) has attracted substantial attention in recent years, which aims to calibrate quantized network $Q$ derived from full-precision network $P$ with no original training samples available. Among the previous efforts, such as GDFQ Xu et al. (2020), the basic idea is to train the generative models Goodfellow et al. (2014); Odena et al. (2017) to produce high-quality samples with various strategies to optimize $Q$, enabling the synthetic samples to asymptotically approximate the real data distribution via distribution reconstruction optimization, as depicted in Fig.2(a), so as to be well distilled from $P$ to $Q$ Hinton et al. (2015); Jin et al. (2023); Li et al. (2022); Hu et al. (2023); Sun et al. (2024).

To address this problem, a large number of methods have emerged, such as generator architecture search Zhu et al. (2021), boundary sample generation Choi et al. (2021), adversarial training Liu et al. (2021) and calibration process optimization Choi et al. (2022). However, such generator-reliance arts generally demand sustained synthesis of numerous samples, while the fidelity of synthetic samples reconstructing real data distribution is inherently constrained by the representational power of generator. Recent efforts Zhong et al. (2022); Zhang et al. (2021); Li et al. (2023) formulate

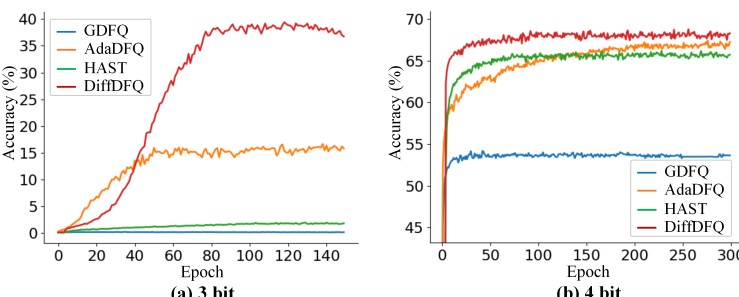

Figure 1: Existing arts, *e.g.*, GDFQ Xu et al. (2020) (the blue) and HAST Li et al. (2023) (the green), bear much larger accuracy loss under (a) 3-bit case than (b) 4-bit case, since the reconstructed samples merely benefit $Q$ with specific bit widths (*e.g.*, 4 bits), not adaptive to 3-bit $Q$; while AdaDFQ Qian et al. (2023a) (the orange) fails to adequately account for the reconstruction fidelity of synthetic samples when adapting to 3-bit case, thereby exhibiting a substantial performance gap under 3-bit case; instead, 4-bit $Q$ well captures reconstructed distribution but bears non-ideal performance, owing to its weaker adaptability than 3-bit $Q$. Our **DiffDFQ** (the red) enjoys the desirable samples by rectifying distribution reconstruction via sample adaptability with forward diffusion process (Eq.(9)).

sample synthesis process as a noise optimization paradigm with *no* generators, where the random noise is iteratively transformed into synthetic samples to approximate the real data distribution, which achieves considerable performance with much less synthetic samples than generator-reliance arts Xu et al. (2020); Choi et al. (2021) (5.12K *vs* 1.20M) for $Q$ over ImageNet for classification (***see Appendix A.1 for more intuitions***). Following this principle, Zhang et al. (2021); Zhong et al. (2022) further improve the diversity and intra-class heterogeneity of synthetic samples, while HAST Li et al. (2023) employs a reweighting strategy to amplify sample difficulty, thereby facilitating the generation of hard samples. Despite of superior distribution reconstruction fidelity for real data samples, these approaches suffer from the non-ideal gap (*e.g.*, the classification output gap) between $P$ and $Q$ with varied bits. Upon them, as illustrated in Fig.2(b), AdaSG Qian et al. (2023b) and AdaDFQ Qian et al. (2023a) focus on the adaptability of synthetic samples (measured by the classification output gap between $P$ and $Q$) to $Q$ with varied bit widths, especially for low-bit cases, and propose a zero-sum game optimization that adaptively modulates the classification output gap between $P$ and $Q$, which, however, impairs the distribution reconstruction of synthetic samples.

In light of the above intuitions, we revisit distribution reconstruction in the DFQ framework via sample adaptability, which leads us to the following observations:

(1) ***The reconstructed samples are merely adaptive to $Q$ with specific bit widths, rather than all bit widths, especially ultra-low bit widths***. Among all $Q$ with varied bit widths, the reconstructed samples from $P$ are actually adaptive to only one, *i.e.*, optimal $Q^*$, benefiting from the desirable gap between $P$ and $Q^*$, which therefore deviate from $Q$ with other bit widths, as indicated in Eq.(6). As a running example, HAST Li et al. (2023) exhibits significantly greater accuracy degradation under 3-bit case compared to its 4-bit counterpart (see Fig.1), since the reconstructed samples are not adaptive to 3-bit $Q$, which is also inferior to AdaDFQ, owing to the adaptability to 3-bit case;

(2) ***AdaDFQ Qian et al. (2023a) achieves the enhanced adaptability across varied bit widths, but compromises the fidelity of original reconstructed information for $Q^*$***. In particular, AdaDFQ overlooks the distribution of reconstructed samples when implementing the adaptability to 3-bit case, consequently bearing a substantial performance gap under 3-bit case (see Fig.1); as opposed, 4-bit $Q$ well captures reconstructed distribution but still bears non-ideal performance, owing to its weaker adaptability than 3-bit $Q$, such as *easy* samples, which, as validated by AdaDFQ, contribute *less* to the performance improvement of *high-bit $Q$*.

The above observations motivate us to delve into *how to rectify distribution reconstruction information via sample adaptability, yielding the desirable samples for $Q$ with varied bit widths?* To this end, we propose a novel Diffusion-Guided Data-Free Quantization approach, dubbed **DiffDFQ**, revealing that: 1) the *principles* of desirable samples for DFQ scheme is to strike an effective balance reconstruction and adaptability information (refer to Eq.(6)); beyond that, 2) the forward diffusion process is considered as an optimal noise strategy to rectify reconstructed samples for obtaining desirable samples, maintaining desirable gap between $P$ and $Q$ with varied bits (refer to Eq.(7) and Eq.(9)). Instead of conventional direct optimization, we devise a *progressive* approximation strategy

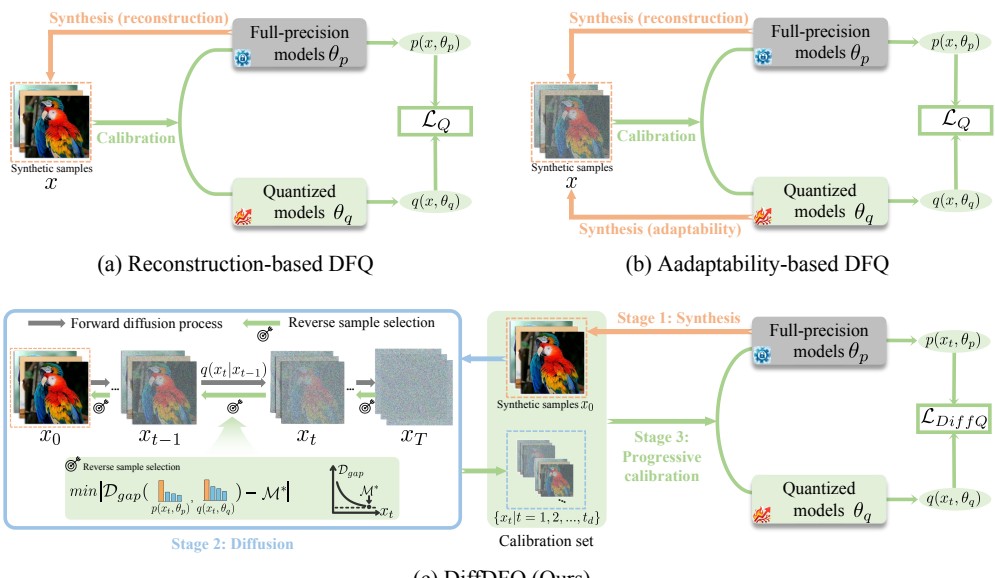

Figure 2: (a) Reconstruction-based DFQ, *e.g.*, Li et al. (2023). (b) Adaptability-based DFQ, *e.g.*, Qian et al. (2023a). (c) Overall architecture of our DiffDFQ, including three stages: 1) *sample synthesis* obtains synthetic samples preserving comprehensive distribution information; 2) *sample diffusion* adds the optimal noise perturbation to approximate a desirable gap $\mathcal{M}^*$; 3) *network calibration* optimizes $Q$ with progressive sample selection over calibration set.

to adaptively perturb reconstructed samples via forward diffusion process regarding the desirable gap between $P$ and $Q$. Technically, we revisit DFQ process from three stages: *sample synthesis* by obtaining samples aimed at preserving complete distribution reconstruction; *sample diffusion* by employing progressive noise infusion to perturb reconstructed samples through *forward diffusion process*; and *network calibration* to optimize $Q$ with progressive selection strategy over a series of diffused samples. **DiffDFQ** conveys the following key contributions and insights: 1) the diffused samples achieve effective balance between sample adaptability and distribution reconstruction for varied $Q$, compared to existing arts, *e.g.*, GDFQ, HAST and AdaDFQ, as illustrated in Fig.1; 2) unlike the generator-reliance arts requiring up to 1.2M synthetic samples, **DiffDFQ** synthesizes merely 5.12K (1K) samples to earn the performance gain over ImageNet for classification.

One may wonder why not relying on *denoising process* regarding the "denoised" samples within reverse time step during forward diffusion process? To kick off the denoising process, one specific optimization target has to be imposed upon diffusion process; it is straightforward to impose the maximum likelihood optimization upon the joint distribution of *fixed* ground-truth sample distribution and latent samples sequence with desirable adaptability. However, as per AdaDFQ Qian et al. (2023a), it is quite tough to optimize the sample adaptability without a generator; worse still, it is quite challenging as weakened distribution reconstruction to prevent the two goals to be simultaneously achieved. Hence, we merely resort to the diffusion process.

It is noteable that **DiffDFQ** performs upon the synthetic samples by generating from the initial Gaussian noise via varied noise optimization strategies Zhong et al. (2022); Li et al. (2023); Jeon et al. (2023), to balance adaptability and reconstruction information for $Q$ with varied bit widths. We empirically validate the merits of **DiffDFQ** as a plug-in module upon these backbones to achieve performance gain to the state-of-the-arts for classification over benchmarks.

## 2 DIFFUSION-GUIDED DATA-FREE QUANTIZATION

### 2.1 PRELIMINARIES: DATA-FREE QUANTIZATION

To facilitate the understanding, we articulate the preliminaries regarding noise optimization based data-free quantization approaches Zhong et al. (2022), which generally consist of the sample synthesis and network calibration process, merely upon the initial pure noise without training samples.

**Sample Synthesis**. Given a series of N initial Gaussian noise $x = \{x^i\}_{i=1}^N$, the synthetic samples are obtained by optimizing these noise to match the real-data distribution serving as the training data to full precision network $P$, which is generally achieved by aligning the batch normalization statistics (BNS) Cai et al. (2020) below:

$$\mathcal{L}_{BNS} = \sum_{l=1}^{L}(||\tilde{\mu}_{p,l}(x) - \mu_l||_2^2 + ||\tilde{\sigma}_{p,l}(x) - \sigma_l||_2^2), \tag{1}$$

where $\tilde{\mu}_{p,l}(x)/\tilde{\sigma}_{p,l}(x)$ and $\mu_l/\sigma_l$ denote the mean/variance of the synthetic samples' distribution within a single batch and the corresponding mean/variance parameters for $P$ at the $l$-th BN layer of the total $L$ layers. Based on that, the inception loss is incorporated to synthesize category-oriented samples as

$$\mathcal{L}_{IL} = \frac{1}{N}\sum_{i=1}^{N}\mathcal{L}_{CE}(p(x^i, \theta_p), y^i), \tag{2}$$

where $\mathcal{L}_{CE}(\cdot, \cdot)$ denotes the Cross-Entropy (CE) loss; $y^i$ is the class label corresponding to $x^i$. $p(x^i, \theta_p)$ is the probability output of a network given the input $x_i$ coupled with $\theta_p$ as the parameter for $P$. Then, the synthetic samples are yielded by optimizing the following objective function:

$$\mathcal{L}_G = \mathcal{L}_{BNS} + \alpha\mathcal{L}_{IL}, \tag{3}$$

where $\alpha$ stands for the balance parameter.

**Network Calibration**. With the off-the-shelf synthetic samples, the quantized network $Q$ is calibrated via the knowledge distillation strategy as per Cross-Entropy (CE) and Kullback-Leibler (KL) loss Zhong et al. (2022), which is formulated as

$$\mathcal{L}_Q = \frac{1}{N}\sum_{i=1}^{N}\mathcal{L}_{CE}(p(x^i, \theta_q), y^i) + \beta\mathcal{L}_{KL}(p(x^i, \theta_p), p(x^i, \theta_q)), \tag{4}$$

where $\beta$ is the balance parameter; $Q$ is parameterized by $\theta_q$. $\mathcal{L}_{KL}(\cdot, \cdot)$ denotes KL divergence loss.

### 2.2 REVISITING DISTRIBUTION RECONSTRUCTION VIA SAMPLE ADAPTABILITY

#### 2.2.1 PROBLEM FORMULATION

As aforementioned in Sec.1, the reconstructed samples from $P$ are usually adaptive to $Q$ with *specific* bit widths, rather than *all* bit widths, owing to the non-ideal gap between $P$ and $Q$ incurred by the quantization error, where the gap between $P$ and $Q$ can be formulated via the following:

$$\mathcal{M}(x) = \mathcal{D}_{gap}(p(x, \theta_p), q(x, \theta_q)), \tag{5}$$

where $\mathcal{D}_{gap}(\cdot, \cdot)$ denotes the metric to measure the difference between the classification outputs (*i.e.*, $p(x, \theta_p)$ and $p(x, \theta_q)$) from $P$ and $Q$, *e.g.*, KL divergence and $\ell$ norm Liu et al. (2021). Following the above, we revisit distribution reconstruction via sample adaptability regarding the gap between $P$ and $Q$ with varied bits, to provide further insights on *what constitutes the desirable samples upon Eq.(5) for $Q$ with varied bit widths in DFQ scheme*. Previous researches Mirzadeh et al. (2020); Wang et al. (2024) on the non-ideal gap issue between $P$ and $Q$, believe that better teacher can't necessarily yield better student, implying that a desirable gap between $P$ and $Q$ is expected given the synthetic samples $x$ as the input to both $P$ and $Q$. *In other words*, with fixed $P$ and $x$, there exists a optimal quantized model $Q^*(x)$ parameterized by $\theta_q^*(x)$, to enjoy the desirable gap $\mathcal{M}^* = \mathcal{D}_{gap}(p(x, \theta_p), p(x, \theta_q^*))$ between $P$ and $Q^*(x)$ for the best network calibration. However, $Q$ with *low bits* generally *differs* greatly from $Q^*(x)$ given $x$, causing the unexpected gap between $P$ and $Q$ to deviate from the desirable gap, which can be characterized via the following *calibration utility*.

**Definition 1** (Calibration Utility). The calibration utility of synthetic samples $x$ for calibrating $Q$ is defined as the proximity of current gap $\mathcal{M}(x)$ to a desirable gap value $\mathcal{M}^*$:

$$Uti(x) = |\mathcal{M}(x) - \mathcal{M}^*|, \tag{6}$$

where $|\cdot|$ returns the absolute value. A smaller value of $Uti(x)$ indicates that the synthetic samples $x$ are more beneficial for $Q$'s calibration.

**Distribution Reconstruction via Adaptability**. In Definition 1, $\mathcal{M}^*$ is not a fixed prior but a theoretical construct determined jointly by $Q$'s inherent capabilities and the calibration objectives, which

is constrained by two aspects of synthetic samples: 1) *Desirable adaptability for $Q$*. The quantization operation is inherently a lossy process due to introduced quantization error. Even with perfect calibration, $Q$ cannot fully replicate all behaviors of $P$. Thus, there exists a theoretically minimal gap $\mathcal{M}_{\min}$, governed by *varied bit widths*. 2) *Sufficient reconstruction information from $P$*. If $\mathcal{M}^*$ is too large, $x$ has likely deviated from the true data distribution, becoming a meaningless outlier. Hence, there exists a maximum gap $\mathcal{M}_{\max}$, determined by the *original data distribution*. Therefore, $\mathcal{M}^*$ should lie within the interval $[\mathcal{M}_{\min}, \mathcal{M}_{\max}]$. The above fact implies that *the desirable samples enjoy rectified distribution reconstruction via sample adaptability*. Note that, under data-free setting, $\mathcal{M}^*$ cannot be directly computed. Instead, we attempt to rectify the synthetic sample $x$ via controlled noise perturbation $\delta$ such that $\mathcal{M}(x)$ can approximate the unknown $\mathcal{M}^*$.

### 2.2.2 OPTIMAL NOISE STRATEGY

**Insight into Controlled Noise Perturbation**. To understand how to steer $\mathcal{M}$ towards $\mathcal{M}^*$, we analyze the effect of adding a small noise perturbation $\delta$ and the properties of optimal perturbation via the following Proposition 1.

**Proposition 1** (Optimal Perturbation Direction). *For small noise perturbation $\delta$ within radius $\varepsilon$, the optimal perturbation direction is aligned with the gradient $\nabla_x \mathcal{M}(x)$.*

*Proof sketch* (see the detailed proof in Appendix A.3). Under the local linearity assumption, we can express the perturbed model discrepancy $\mathcal{M}(x + \delta)$ using a first-order Taylor expansion:

$$\mathcal{M}(x + \delta) \approx \mathcal{M}(x) + \nabla_x \mathcal{M}(x)^T \delta, ||\delta|| \leq \varepsilon. \tag{7}$$

By the Cauchy–Schwarz inequality, we have $|\nabla_x \mathcal{M}(x)^T \delta| \leq ||\nabla_x \mathcal{M}(x)|| \cdot ||\delta|| \leq \varepsilon ||\nabla_x \mathcal{M}(x)||$, where the equality holds if and only if $\delta$ is linearly correlated with $\nabla_x \mathcal{M}(x)$ and maintains the same direction. Therefore, the optimal perturbation is $\delta^* = \varepsilon \cdot \frac{\nabla_x \mathcal{M}(x)}{||\nabla_x \mathcal{M}(x)||}$. $\qquad\square$

Proposition 1 implies that the maximum of the inner product $\nabla_x \mathcal{M}(x)^T \delta$ is achieved when $\delta$ is aligned with the gradient $\nabla_x \mathcal{M}(x)$; it is the minimum otherwise. Therefore, by carefully modulating both the direction and magnitude of the injected noise $\delta$, we can precisely increase or decrease the value of $\mathcal{M}(x)$ towards the optimal value $\mathcal{M}^*$, yielding desirable samples for varied bit widths.

**Forward Diffusion Process as Optimal Noise Strategy**. Despite the optimality, it is required to compute the gradient for every synthetic sample, which is computationally prohibitive. Therefore, we require a practical strategy that approximately follows the gradient's guidance to preserve semantic coherence while enables the adaptability for varied $Q$. To facilitate this insight, we define the following forward diffusion process in diffusion models.

**Definition 2** (Forward Diffusion Process). Given the initial samples $x_0$, the forward diffusion process is specialized as *progressively* adding the random noise with the variance schedule $\beta_t$ (*see Appendix A.8.5 for more intuitions*), to yield the noisy samples $x_t$ within the $t \in \{0, 1, ..., T\}$-th time step from the total $T$ time steps (see Fig.2(c)), which is defined as a Markov chain:

$$q(x_t|x_0) := \mathcal{N}(x_t; \sqrt{\bar{\alpha}_t}x_0, (1 - \bar{\alpha}_t)\mathbf{I}), \tag{8}$$

where $\sqrt{\bar{\alpha}_t} := \prod_{s=0}^{t} \alpha_s$ and $\alpha_s = 1 - \beta_s$. Hence, the closed-form expression for any timestep $t$ is a linear combination of $x_0$ and noise variable $\epsilon_t$:

$$x_t = \sqrt{\bar{\alpha}_t}x_0 + \sqrt{1 - \bar{\alpha}_t}\epsilon_t, \epsilon_t \in \mathcal{N}(\mathbf{0}, \mathbf{I}). \tag{9}$$

Revisiting Eq.(9), we observe that the first term ensures a structured decay of the reconstructed samples $x_0$ that preserves distribution reconstruction information; while the second term introduces precisely controlled perturbation through the Gaussian noise $\epsilon_t$, to fulfill desirable adaptability to varied bit widths. The inherent continuity and structure of forward diffusion process naturally give rise to the following Proposition 2.

**Proposition 2** (Optimal Noise Strategy). *The forward diffusion process is an optimal and efficient strategy for obtaining perturbed sample $x_t$ to minimize $|\mathcal{M}(x_t) - \mathcal{M}^*|$.*

*Proof sketch* (see the detailed proof in Appendix A.3). We justify this proposition from the following three aspects: 1) *Controllability*. The time step $t$ acts as a continuous control parameter for optimizing $\mathcal{M}(x_t)$ towards $\mathcal{M}^*$. 2) *Semantic consistency*. The fine-grained noise schedule $\beta_t$ ensures minimal perturbations (e.g., from $10^{-4}$ to $0.02$) to satisfy local linearity assumption. 3) *Equivalence of Gaussian perturbation*. Gaussian noise automatically induces loss changes proportional to $||\nabla_x \mathcal{M}(x)||^2$, without computing per-sample gradient. $\qquad\square$

The Proposition 2 paves the way for our proposed Diffusion-Guided Data-Free Quantization approach, dubbed **DiffDFQ**, as illustrated in Fig.2(c), which decomposes DFQ process into three stages: *sample synthesis* to synthesize deterministic samples, *i.e.*, $x$, preserving comprehensive distribution information (Sec.2.1); *sample diffusion* to indirectly approximate desirable samples for varied $Q$ by progressively infusing the noise to $x$ through forward diffusion process (Sec.2.3); and *network calibration* to optimize $Q$ with progressive selection over diffused samples (Sec.2.4).

### 2.3 SAMPLE DIFFUSION: DISTRIBUTION RECONSTRUCTION VIA DIFFUSION PROCESS

To fulfill desirable samples for varied $Q$, we progressively add the noise perturbation to the reconstructed samples $x$ from the sample synthesis stage through the forward diffusion process. Formally, given $x_0 = x$, a series of diffused samples are obtained via the noise schedule Ho et al. (2020) below:

$$x_t = Diff(x, t), t = 1, 2, ..., T, \quad (10)$$

where $Diff(\cdot, \cdot)$ denotes the sample diffusion function based on Eq.(9).

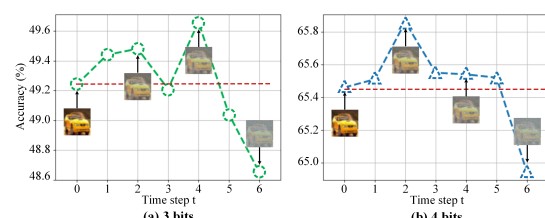

(a) 3 bits      (b) 4 bits

Figure 3: Motivation experiments for classification with varied $Q$: (a) 3 bits and (b) 4 bits on CIFAR-100. The diffused samples within a certain range of early time steps exhibit better adaptability to $Q$ than original synthetic samples ($x_0$).

**Not all diffused samples are adaptive to $Q$.** To understand the above intuitions, we conduct $Q$'s calibration by substituting the original synthetic samples $x$ with one of the diffused samples $x_t$ (see Fig.3 and Appendix A.4), which reveals the followings: *the diffused samples within a certain range (e.g., $x_5, x_4, ..., x_0$) of the early time steps enjoy better balance between distribution reconstruction and adaptability than $x$, thus approaching the desirable samples*; *the desirable samples should dynamically progress from $x_t$ to $x_0$ as $Q$ is updated, since updating $Q$ requires more reconstruction information to improve itself*. The above fact implies that the diffusion process extends the adaptability of $x$ to varied $Q$, while ensemble of diffused samples from varied time steps further balances the reconstruction information when updating $Q$, which naturally motivates the progressive calibration in the next.

### 2.4 PROGRESSIVE CALIBRATION: OPTIMIZING $Q$ WITH PROGRESSIVE SAMPLE SELECTION

As previously discussed, according to the time step $t_d$ (discussed in Appendix A.8.1 and A.4) for diffusion process, a series of diffused samples $x_1, ..., x_{t_d}$, together with original synthetic samples $x_0$, constitute the calibration set to optimize $Q$. To yield desirable samples for updating $Q$, we devise a progressive sample selection strategy, which progressively selects proper diffused samples $x_t$ as per the order from $t = t_d$ to 0, as observed in Fig.4, involving the following selection strategies: (1) **Uniform selection** performs by uniformly selecting the diffused



Figure 4: Illustration of progressive calibration process over synthetic samples and diffused samples, which allocates proper number of epochs (*e.g.*, $E_{t_d}$, $E_{t_d-1}$ and $E_0$) via varied selection strategy.

samples at each time step with equal epoch interval, *e.g.*, $E_{t_d} = E_{t_d-1} = ... = E_0$. Such strategy actually depends on the assumption that the samples from different time steps maintain the equal contribution to $Q$ at each training state. (2) **Non-uniform selection** equips varied training epochs for the diffused samples from different time steps, *e.g.*, $E_{t_d} < E_{t_d-1} < ... < E_0$, since the diffused samples from large time steps actually contain little semantics with more noise as degraded, which may contribute less to the calibration of $Q$. (3) **Adaptive selection** dynamically selects the diffused samples according to the convergence of $Q$. Notably, such adaptive strategy is essentially a special non-uniform strategy. Practically, we select the combination of these strategies with $Q$ varied; for instance, we prefer adaptive strategy for low-bit $Q$, since less noise degraded for smaller time steps easily leads to stable convergence, while non-uniform strategy may be better for high-bit **Q**. Finally, for each $x_t$ at time step $t$, the calibration loss for $Q$ is formulated as

$$\mathcal{L}_{DiffQ} = \frac{1}{N} \sum_{i=1}^{N} \mathcal{L}_{CE}(p(x_t^i, \theta_q), y^i) + \beta \mathcal{L}_{KL}(p(x_t^i, \theta_p), p(x_t^i, \theta_q)). \quad (11)$$

With Eq.(11) to be minimized, $Q$ is updated to recover the performance under rapid convergence (as discussed in Sec.3.5). It is noteworthy that Eq.(11) is the optimization target to calibrate $Q$, based on

the diffused samples obtained from progressive diffusion process, which, as aforementioned, progressively diffuses the samples with no deterministic optimization target (see the algorithm details and validation experiments in Appendix A.5 and A.6).

# 3 EXPERIMENT

## 3.1 EXPERIMENTAL SETTINGS AND DETAILS

To evaluate **DiffDFQ**, we conduct the experiments on three typical image classification datasets, including: **CIFAR-10** and **CIFAR-100** Krizhevsky (2009) contain 10 and 100 classes of images, which are split into 50K images for training and 10K images for testing; **ImageNet** (**ILSVRC2012**) Russakovsky et al. (2015) consists of 1.2M training images and 50k validation images from 1000 categories. For DFQ, only validation sets are adopted to evaluate the quantized models ($Q$). We quantize pre-trained full-precision networks ($P$) including ResNet-20 for CIFAR, and ResNet-18 He et al. (2016), ResNet-50 and MobileNetV2 Sandler et al. (2018) for ImageNet, via the following quantizer to yield $Q$:

**Quantizer.** Following Xu et al. (2020); Choi et al. (2021), we quantize both full-precision (float32) weights and activations into $n$-bit precision by a symmetric linear quantization method as Jacob et al. (2018):

$$\theta_q = round\Big((2^n - 1) * \frac{\theta_p - \theta_{min}}{\theta_{max} - \theta_{min}} - 2^{n-1}\Big), \tag{12}$$

where $\theta_p$ and $\theta_q$ are the full-precision and quantized value; $round(\cdot)$ returns the nearest integer value to the input. $\theta_{min}$ and $\theta_{max}$ are the minimum and maximum of $\theta_p$. For *sample synthesis*, 5120 (or 1000) synthetic samples are optimized via Eq.(3) using Adam Kingma & Ba (2014) optimizer with a momentum of 0.9 and an initial learning rate of 0.5. The batch size is set as 256. For *sample diffusion*, the linear noise schedule ($\beta_t$) is adopted with a total of $T = 80$ time steps. For *network calibration*, $Q$ is optimized via Eq.(11) using SGD optimizer with a momentum of 0.9 and the weight decay of $10^{-4}$. The batch size for calibration process is set as 256 for CIFAR-10/100 and 16 for ImageNet. The initial learning rate is set as $10^{-5}$ and $10^{-6}$ for CIFAR-10/100 and ImageNet. The learning rates are decayed by 0.1 every 100 epochs with a total of 400 epochs. For hyper-parameters, $\alpha$ in Eq.(3) and $\beta$ in Eq.(11), are set as 1 and 20. All experiments are implemented with pytorch Paszke et al. (2019) via the code of **IntraQ** Zhong et al. (2022), **HAST** Li et al. (2023) and **GENIE** Jeon et al. (2023), and run on an NVIDIA GeForce GTX 1080 Ti GPU and an Intel(R) Core(TM) i7-6950X CPU @ 3.00GHz. ***More discussions about hyperparameters, noise schedules, time steps and sample number are accessible in Appendix A.8.***

## 3.2 COMPARISON WITH STATE-OF-THE-ARTS

To verify the superiority of **DiffDFQ**, we compare it with the following typical DFQ arts: 1) quantization-aware training (**QAT**) approaches: **GDFQ** Xu et al. (2020), **Qimera** Choi et al. (2021) and **RIS** Bai et al. (2024) utilize a generator to reconstruct the original data; **AdaSG** Qian et al. (2023b) and **AdaDFQ** Qian et al. (2023a) focus on the sample adaptability; **IntraQ** Zhong et al. (2022), **HAST** Li et al. (2023) and **PLF** Fan et al. (2024) treat the sample synthesis as the problem of noise optimization with no generator; **TexQ** Chen et al. (2023) aims to retain detailed texture distribution in the real samples; **SYNQ** Kim et al. (2025) attempts to overcome the limitations of fune-tuning $Q$ with synthetic samples; **Enhancing** Zhao et al. (2025) enhances both inter-class and intra-class diversity; 2) post-training quantization (**PTQ**) approaches: **DSG** Qin et al. (2023) focuses the diversity of synthetic samples; **GENIE** Jeon et al. (2023) simultaneously learns a generator and its inputs initialized from a Gaussian distribution to construct the synthetic samples; **SADAG** Dung et al. (2024) considers quantized model sharpness in sample synthesis. In particular, **DiffDFQ** works by rectifying reconstruction information of the off-the-shelf reconstructed samples from existing arts, *i.e.*, **IntraQ**, **HAST** and **GENIE**, denoted as **IntraQ+DiffDFQ**, **HAST+DiffDFQ** and **GENIE+DiffDFQ**. *Notably*, we adopt the non-uniform strategy for high-bit $Q$ (*e.g.*, 4 bits and 5 bits) and adaptive strategy for low-bit $Q$ (*e.g.*, 2 bits and 3 bits) in most cases, while few cases prefer the uniform strategy.

Tab.1 summarize our findings below: 1) **DiffDFQ** achieves a significant and consistent accuracy gain over the state-of-the-arts for $Q$ under varied bit-widths, confirming the intuition of rectifying the distribution reconstruction of off-the-shelf synthetic samples with no generator to derive the desirable samples for varied $Q$ via the diffusion process. Specifically, **DiffDFQ** receives at most

Table 1: Classification accuracy (%) comparison with the state-of-the-arts on ImageNet. -: no results are reported (no validation on specific dataset). †: the results reproduced by author-provided code. $m$w$n$a indicates the weights and activations are quantized to $m$-bit and $n$-bit. #Data: the number of synthetic samples. The best results are reported with **boldface**.

| Model | #Data | ResNet-18 (71.47) | | | | | ResNet-50 (77.73) | | | | | MobileNet-V2 (73.03) | | | | |
|---|---|---|---|---|---|---|---|---|---|---|---|---|---|---|---|---|
| Bit width ($n$w$n$a) | | 2w2a | 2w4a | 3w3a | 4w4a | 5w5a | 2w2a | 2w4a | 3w3a | 4w4a | 5w5a | 2w2a | 2w4a | 3w3a | 4w4a | 5w5a |
| **Data-Free Quantization-Aware Training (QAT)** | | | | | | | | | | | | | | | | |
| **GDFQ** Xu et al. (2020) (ECCV) | 1.20M | - | - | 20.23 | 60.60 | 68.49 | - | - | - | 54.16 | 71.63 | - | - | - | 59.43 | 68.11 |
| **Qimera** Choi et al. (2021) (NeurIPS) | 1.20M | - | - | - | 63.84 | 69.29 | - | - | - | 66.25 | 75.32 | - | - | - | 61.62 | 70.45 |
| **AdaSG** Qian et al. (2023b) (AAAI) | 1.20M | - | - | 37.04 | 66.50 | 70.29 | - | - | 16.98 | 68.58 | 76.03 | - | - | 26.90 | 65.15 | 71.61 |
| **AdaDFQ** Qian et al. (2023a) (CVPR) | 1.20M | - | - | 38.10 | 66.53 | 70.29 | - | - | 17.63 | 68.38 | 76.03 | - | - | 28.99 | 65.41 | 71.61 |
| **TexQ** Chen et al. (2023) (NeurIPS) | 1.20M | - | - | 50.28 | 67.73 | - | - | - | 25.27 | 70.72 | - | - | - | 32.80 | 67.07 | - |
| **AIT+RIS** Bai et al. (2024) (AAAI) | 1.20M | - | - | - | 67.55 | 70.59 | - | - | - | 71.54 | 76.36 | - | - | - | 67.61 | **72.05** |
| **Enhancing** Zhao et al. (2025) (CVPR) | 1.20M | - | - | 52.27 | 68.13 | - | - | - | 28.32 | 71.20 | - | - | - | 34.10 | 67.10 | - |
| **PLF** Fan et al. (2024) (CVPR) | 5.12K | - | - | - | 67.02 | 70.35 | - | - | - | 68.97 | 76.08 | - | - | - | - | - |
| **SYNQ** Kim et al. (2025) (ICLR) | 5.12K | - | - | 52.02 | 67.90 | - | - | - | 26.89 | 71.05 | - | - | - | 34.21 | 67.27 | - |
| **IntraQ** Zhong et al. (2022) (CVPR) | 5.12K | - | - | 41.46† | 66.47 | 69.94 | - | - | 0.18† | 64.19† | 74.43† | - | - | 14.84† | 65.10 | 71.28 |
| **IntraQ+DiffDFQ (Ours)** | 5.12K | - | - | 44.46 ±0.63 | 66.26 ±0.06 | 69.96 ±0.02 | - | - | 8.37 ±0.37 | 64.59 ±0.20 | 74.71 ±0.07 | - | - | 17.23 ±0.34 | 65.50 ±0.10 | 71.54 ±0.07 |
| **Gains** | - | - | - | +3.00 | -0.21 | +0.02 | - | - | +8.29 | +0.40 | +0.28 | - | - | +2.39 | +0.40 | +0.26 |
| **HAST** Li et al. (2023) (CVPR) | 5.12K | - | - | 48.44† | 66.47† | - | - | - | 1.98† | 66.17† | 75.55† | - | - | 32.48† | 63.02† | 70.37† |
| **HAST+DiffDFQ (Ours)** | 5.12K | - | - | 49.78 ±0.12 | 66.64 ±0.05 | - | - | - | 38.37 ±1.05 | 68.78 ±0.88 | 75.80 ±0.08 | - | - | 33.97 ±0.50 | 63.76 ±0.08 | 70.70 ±0.13 |
| **Gains** | - | - | - | +1.34 | +0.17 | - | - | - | +36.39 | +2.61 | +0.25 | - | - | +1.49 | +0.74 | +0.33 |
| **Data-Free Post-Training Quantization (PTQ)** | | | | | | | | | | | | | | | | |
| **DSG** Qin et al. (2023) (T-PAMI) | 1K | - | - | - | 66.67 | - | - | - | - | 68.30 | - | - | - | - | - | - |
| **SADAG** Dung et al. (2024) (ICML) | 1K | 54.51 | 65.25 | 67.10 | 69.72 | - | **57.55** | 70.52 | 72.62 | 75.70 | - | 13.01 | 51.89 | 56.02 | 68.54 | - |
| **GENIE** Jeon et al. (2023) (CVPR) | 1K | 53.74 | 65.10 | 66.89 | 69.66 | 70.53† | 56.81 | 69.99 | 72.54 | 75.59 | 76.36† | 11.93 | 51.47 | 55.31 | 68.38 | 71.48† |
| **GENIE+DiffDFQ (Ours)** | 1K | **54.60** ±0.06 | **65.30** ±0.11 | **67.14** ±0.03 | **69.77** ±0.03 | 70.68 ±0.05 | 56.72 ±0.05 | 70.63 ±0.20 | 72.65 ±0.07 | 75.70 ±0.02 | 76.43 ±0.03 | 13.04 ±0.02 | 51.95 ±0.04 | 56.15 ±0.07 | 68.62 ±0.04 | 71.61 ±0.03 |
| **Gains** | - | +0.86 | +0.20 | +0.25 | +0.11 | +0.15 | -0.09 | +0.64 | +0.11 | +0.11 | +0.07 | +1.11 | +0.47 | +0.84 | +0.24 | +0.13 |

36.39% accuracy gains on ImageNet over **IntraQ**, **HAST** and **GENIE**. In particular, **GENIE** outperforms **IntraQ**, owing to its focus on the optimization of quantization algorithm. 2) **DiffDFQ** exhibits considerable advantages over the generator-reliance DFQ arts, where the massive generated samples are required via generators. Notably, compared with **GDFQ**, **Qimera**, **AdaSG**, **AdaDFQ**, **TexQ**, **RIS** and **Enhancing**, **DiffDFQ** still earns an accuracy gain with a large margin (at most 55.67% on ImageNet) with few synthetic samples, implying the *benefits* of unlocking the sample adaptability from limited number of off-the-shelf reconstructed samples with neither generator nor sample adaptability optimization via the diffusion process, to be in line with the intuition in Sec.2.2. 3) **DiffDFQ** delivers the improvements for $Q$ with ultra low bits, verifying the *importance* of rectifying reconstruction and adaptability information for $Q$ varied bit widths. As reported in Tab.1, most of the methods failed on 2 bits with even *no* results; for 2w2a and 2w4a, **DiffDFQ** exhibits the obvious accuracy advantages at most 1.11% over **GENIE** (*see more results in Appendix A.7*).

## 3.3 ABLATION STUDIES

### 3.3.1 DISCUSSION ON EACH COMPONENT OF DIFFDFQ

We further ablate **DiffDFQ** to show the necessity of following parts, *i.e.*, progressive calibration ($\mathcal{L}_{DiffQ}$ in Eq.(11)) with diffused samples and sample selection strategies (Fig.4) over diffused samples. Specifically, we substitute $\mathcal{L}_{DiffQ}$ with $\mathcal{L}_Q$ in Eq.(4) by utilizing the diffused samples from only one time step, denoted as *w/o* $\mathcal{L}_{DiffQ}$; and abandon the sample selection strategies to adopt random selection strategy instead, denoted as *w/o* progressive selection, where the ablation experiments are conducted with IntraQ+DiffDFQ on CIFAR-100. Tab.2 suggests the significant superiority (50.44% and 66.09%) of **DiffDFQ** over other cases. Notably, compared to **DiffDFQ**, the cases *w/o* $\mathcal{L}_{DiffQ}$ and *w/o* progressive selection suffer from a sharp accuracy loss (at most 1.16%

Table 2: Ablation study for classification accuracy (%) about varied components of **DiffDFQ** on CIFAR-100. $n$w$n$a indicates the weights and activations are quantized to $n$-bit. The best results are reported with **boldface**.

| Method | 3w3a | 4w4a |
|---|---|---|
| Baseline (*w/o* sample diffusion) | 48.25 | 64.98 |
| *w/o* progressive selection | 49.28 | 65.72 |
| *w/o* $\mathcal{L}_{DiffQ}$ | 49.66 | 65.86 |
| **DiffDFQ (Ours)** | **50.44** | **66.09** |

and at least 0.23%), , implying that the progressive calibration by exploiting *a series of* diffused samples, rather than ones from *single* time step, is beneficial to balancing distribution reconstruction sample adaptability as $Q$ is updated, which, in turn, highlights the *necessity* of the subsequent diffusion process in Sec.2.2 and Sec.2.3.

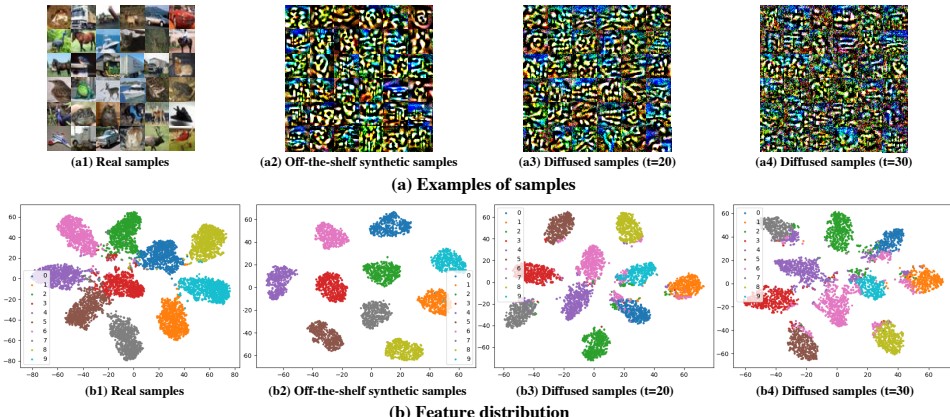

(a1) Real samples    (a2) Off-the-shelf synthetic samples    (a3) Diffused samples (t=20)    (a4) Diffused samples (t=30)

**(a) Examples of samples**

(b1) Real samples    (b2) Off-the-shelf synthetic samples    (b3) Diffused samples (t=20)    (b4) Diffused samples (t=30)

**(b) Feature distribution**

Figure 5: (a) Visualization of real samples, off-the-shelf reconstructed samples and diffused samples, along with (b) feature distribution of the corresponding samples comprising 10 categories (one color refers to one category) using t-SNE Van der Maaten & Hinton (2008) on CIFAR-10. The distribution of the diffused samples is more similar to that of real samples.

### 3.4 VISUAL ANALYSIS AND PRIVACY RISK ON DIFFUSED SAMPLES

To further show the merits of **DiffDFQ**, we perform the visual analysis on the off-the-shelf synthetic samples and the corresponding diffused samples from varied time steps. Fig.5(a) shows the examples of real samples in Fig.5(a1), off-the-shelf synthetic samples in Fig.5(a2) and the diffused samples from the time step $t = 20$ in Fig.5(a3) and $t = 30$ in Fig.5(a4); while Fig.5(b) illustrates the feature distribution of the corresponding samples using t-SNE Van der Maaten & Hinton (2008) on CIFAR-10. As observed in Fig.5(b3)(b4), the distribution of the diffused samples is more similar to that of real samples in Fig.5(b1) compared to the synthetic samples in Fig.5(b2), *implying* that **DiffDFQ** delivers the sample adaptability information as expected and extra meaning information (*e.g.*, the hardness and diversity) over the distribution reconstruction information of synthetic samples by diffusing the synthetic samples, which is in line with Sec.2.2. Meantime, the unrealistic synthetic samples in Fig.5(a3)(a4) also avoid privacy leakage.

### 3.5 ANALYSIS ON TRAINING COMPLEXITY

To verify the efficiency of DiffDFQ, we perform the training complexity analysis with ResNet-50 on ImageNet by computing the convergence time (GPU hours) of $Q$ based on HAST and HAST+DiffDFQ, as reported in Tab.3. The results indicate that DiffDFQ converges faster than HAST during the calibration process of $Q$, while the convergence time is reduced by at most 35.5% (18.96h *vs* 12.23h), implying that the diffused samples can better

Table 3: Analysis on training complexity (GPU hours). #Data: the number of synthetic samples. The best results are reported with **boldface**.

| Method | #Data | 3 bits | 4 bits | 5 bits |
|---|---|---|---|---|
| **HAST** Li et al. (2023) | 5.12K | 14.95h | 18.96h | 16.83h |
| **DiffDFQ (Ours)** | 5.12K | **10.06h** | **12.23h** | **14.94h** |

rectify the distribution reconstruction via adaptability, thus facilitating the convergence of $Q$.

## 4 CONCLUSION AND LIMITATIONS

This paper proposes a novel diffusion-guided data-free quantization approach, dubbed **DiffDFQ**, which rectifies distribution information constrained by sample adaptability in a progressive approximation manner, yielding desirable samples for varied $Q$. Specifically, **DiffDFQ** comprises three stages: sample synthesis by obtaining reconstructed samples; upon that, sample diffusion by progressively infusing the noise for yielding desirable samples; and network calibration to calibrate $Q$ with progressive selection over diffused samples. We empirically verify the merits of **DiffDFQ** as a plug-in module on top of varied noise optimization strategies to yield the superior performance, especially ultra-low bit widths, over the state-of-the-arts. We admit that, our model and most of other arts currently involve classification tasks, which can further be extended to other tasks and backbones in future work.

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

# A APPENDIX

## A.1 COMPARISON WITH EXISTING DFQ ARTS IN SEC.1

Data-free quantization (DFQ) approaches aim to calibrate quantized network $Q$ derived from full-precision network $P$ with no training samples available, which generally involve two types of schemes: *generator-based DFQ* and *noise optimization-based DFQ*. Specifically, the basic idea of *generator-based DFQ* is to train the generative models to produce high-quality samples with various strategies to optimize $Q$, enabling the synthetic samples to asymptotically approximate the real data distribution via distribution reconstruction optimization, so as to be well distilled from $P$ to $Q$. However, the generator-based arts generally demands sustained synthesis of numerous samples (see Tab.4), while the fidelity of synthetic samples reconstructing real data distribution is inherently constrained by the representational power of generator. In contrast, *noise optimization-based DFQ* formulates sample synthesis process as a noise optimization paradigm with no generators, where the random noise is iteratively transformed into synthetic samples to approximate the real data distribution, which achieves considerable performance with much less synthetic samples than generator-reliance arts (5.12K vs 1.20M) for $Q$ over ImageNet for classification; see Tab.4.

Table 4: **Comparison with the existing DFQ arts.** **DiffDFQ** achieves considerable performance with much less synthetic samples to Q with 3 bits than others over ImageNet with ResNet-18 for classification. †: the results reproduced by author-provided code. Reconstructed means the synthetic samples with similar distribution as original real samples.

| Scheme | Method | Data type | #Synthetic data | Accuracy (%) |
|---|---|---|---|---|
| Generator-based DFQ | **GDFQ** Xu et al. (2020) | Reconstructed | 1.20M | 20.23 |
| | **Qimera** Choi et al. (2021) | Reconstructed | 1.20M | - |
| | **AdaDFQ** Qian et al. (2023a) | Adaptability to $Q$ | 1.20M | 38.10 |
| | **RIS** Bai et al. (2024) | Reconstructed | 1.20M | - |
| Noise optimization-based DFQ | **IntraQ** Zhong et al. (2022) | Reconstructed | 5.12K | 41.46 |
| | **HAST** Li et al. (2023) | Reconstructed | 5.12K | 48.44† |
| | **GENIE** Jeon et al. (2023) | Reconstructed | 1K | 66.89 |
| Noise optimization-based DFQ | **IntraQ+** DiffDFQ (Ours) | Reconstructed + Adaptability to $Q$ | 5.12K | 44.46±0.63 |
| | **HAST+** DiffDFQ (Ours) | Reconstructed + Adaptability to $Q$ | 5.12K | 49.78±0.12 |
| | **GENIE+** DiffDFQ (Ours) | Reconstructed + Adaptability to $Q$ | 1K | **67.14±0.03** |

## A.2 MORE INTUITIONS ON EQ.(6)

As discussed in Sec.2.2, we revisit the distribution reconstruction via sample adaptability, and establish the potential relationship between desirable sample adaptability and desirable gap; hence, we can rectify the distribution reconstruction of synthetic samples by exploiting the desirable gap between $P$ and $Q$, yielding desirable samples for $Q$ with varied bit widths.

To further show the intuitions, we study the output gap ($\mathcal{M}$ in Eq.(5)) between $P$ and $Q$ with varied bit widths over HAST and DiffDFQ, to reveal the principle of the sample adaptability. Fig.6 illustrates that lower-bit $Q$ possesses much larger gap between $P$ and $Q$, owing to larger quantization error from $P$ to $Q$; while the gap for DiffDFQ is greatly reduced across varied bit widths, especially for 3-bit case, since the distribution reconstruction of synthetic samples from HAST is rectified to approximate the desirable gap, yielding the samples with desirable adaptability to varied $Q$, which implies that the desirable sample adaptability can be achieved by adaptively regulating the gap between $P$ and $Q$ (refer to Eq.(7)). The above further reveals " *for varied $Q$, the desirable gap is expected to well rectify distribution reconstruction constrained by sample adaptability, to yield the desirable samples*".

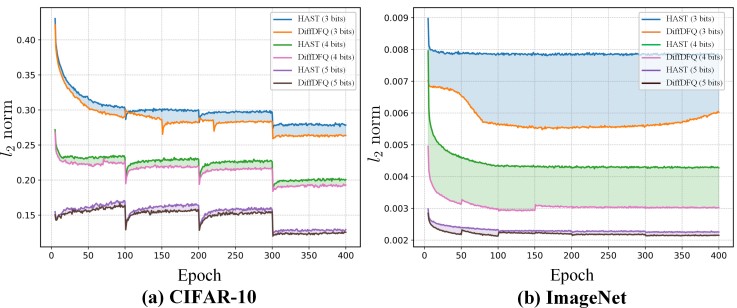

(a) CIFAR-10       (b) ImageNet

Figure 6: Illustration of the desirable gap between P and Q with varied bits facilitating rectifying distribution reconstruction from HAST Li et al. (2023), yielding desirable samples.

### A.3 PROOF

In this section, we further provide the detailed proofs of Proposition 1 and Proposition 2.

**Proposition 1** (Optimal Perturbation Direction). *For small noise perturbation $\delta$ within radius $\varepsilon$, the optimal perturbation direction is aligned with the gradient $\nabla_x \mathcal{M}(x)$.*

*Proof.* We first provide the following assumption as the mathematical foundation for the subsequent optimal perturbation analysis.

**Assumption 1** (Local Linearity). Given synthetic sample $x$, there exists a neighborhood $\mathcal{B}(x, \varepsilon) = \{x' : |x' - x| \leq \varepsilon\}$, where the behaviors of both full-precision and quantized models can be well-approximated by linear functions.

The assumption can be justified from two aspects: 1) modern deep models typically employ piecewise linear activation functions (*e.g.*, ReLU and LeakyReLU) or approximately linear smooth functions (*e.g.*, GELU), which leads to strongly linear behavior within local regions; 2) the training process of neural networks (*e.g.*, SGD) fundamentally depends on local linear approximations of the loss function in parameter space.

Under the local linearity assumption, we can express the perturbed model discrepancy $\mathcal{M}(x + \delta)$ using a first-order Taylor expansion:

$$\mathcal{M}(x + \delta) \approx \mathcal{M}(x) + \nabla_x \mathcal{M}(x)^T \delta, \, \|\delta\| \leq \varepsilon, \tag{13}$$

where the term $\nabla_x \mathcal{M}(x)$ can be decomposed via the chain rule as follows:

$$\begin{aligned} \nabla_x \mathcal{M}(x) &= \left( \frac{\partial \mathcal{M}(x)}{\partial p(x, \theta_p)} \right)^T \frac{\partial p(x, \theta_p)}{\partial x} + \left( \frac{\partial \mathcal{M}(x)}{\partial q(x, \theta_q)} \right)^T \frac{\partial q(x, \theta_q)}{\partial x} \\ &= J_{p(x,\theta_p)}^T \nabla_{p(x,\theta_p)} \mathcal{M}(x) + J_{q(x,\theta_q)}^T \nabla_{q(x,\theta_q)} \mathcal{M}(x), \end{aligned} \tag{14}$$

where $J_{p(x,\theta_p)}$ and $J_{q(x,\theta_q)}$ represent the Jacobian matrices of $p(x, \theta_p)$ and $q(x, \theta_q)$; while $\nabla_{p(x,\theta_p)} \mathcal{M}(x)$ and $\nabla_{q(x,\theta_q)} \mathcal{M}(x)$ denote the gradients of the model discrepancy function with respect to the prediction outputs. Based on Eq.(13), the change in the model discrepancy function $\mathcal{M}(x)$ can be expressed as

$$\Delta \mathcal{M} = \mathcal{M}(x + \delta) - \mathcal{M}(x) \approx \nabla_x \mathcal{M}(x)^T \delta, \tag{15}$$

By the Cauchy–Schwarz inequality, we have

$$|\nabla_x \mathcal{M}(x)^T \delta| \le ||\nabla_x \mathcal{M}(x)|| \cdot ||\delta|| \le \varepsilon ||\nabla_x \mathcal{M}(x)||, \tag{16}$$

where the equality holds if and only if $\delta$ is linearly correlated with $\nabla_x \mathcal{M}(x)$ and maintains the same direction. To yield the maximum change of $\mathcal{M}(x)$, the optimal perturbation can be obtained below:

$$\delta^* = \varepsilon \cdot \frac{\nabla_x \mathcal{M}(x)}{||\nabla_x \mathcal{M}(x)||}. \tag{17}$$

Therefore, the optimal perturbation direction is aligned with the gradient $\nabla_x \mathcal{M}(x)$.

$\square$

**Proposition 2** (Optimal Noise Strategy). *The forward diffusion process is an optimal and efficient strategy for obtaining perturbed sample $x_t$ to minimize $|\mathcal{M}(x_t) - \mathcal{M}^*|$.*

*Proof.* We justify this proposition from the following three parts of forward diffusion process: *controllability*, *semantic consistency* and *equivalence of Gaussian perturbation*.

**Part 1**: Controllability

The timestep $t$ provides a continuous control parameter. As $t$ increases from $0$ to $T$, the effective noise level $\sqrt{1 - \bar{\alpha}_t}$ increases monotonically from 0 to 1. This provides a continuous "knob" for adjusting the perturbation intensity. Empirically, we observe that $\mathcal{M}(x_t)$ is a monotonic function of $t$ for a wide range of values. This allows us to efficiently find the optimal timestep $t^*$ that minimizes our objective through a simple one-dimensional search.

**Part 2**: Semantic consistency

The structure of the diffusion forward process ensures that perturbed samples retain sufficient semantics, which is crucial for effective calibration of $Q$. $\sqrt{\bar{\alpha}_t}$ progressively attenuates the original signal $x_0$ rather than merely adding noise to it. Furthermore, standard noise schedules (*e.g.*, linear and cosine) ensure each step's perturbation $\beta_t$ is small, guaranteeing that the local linearity assumption holds at every microscopic step of the Markov chain. Consequently, the diffused sample $x_t$ can remain on or near the manifold of original synthetic sample $x_0$, thereby the model discrepancy $\mathcal{M}(x_t)$ is informative of the true quantization error. This property enforces the upper bound $\mathcal{M}_{max}$, as discussed in Sec.2.2.

**Part 3**: Equivalence of Gaussian perturbation

While Proposition 1 suggests a deterministic perturbation direction, *i.e.*, $\delta \propto \nabla_x \mathcal{M}$, we employ isotropic Gaussian noise $\epsilon$ in practice, since this stochastic strategy is statistically optimal. Let the noise perturbation $\delta = \sigma\epsilon$, where $\epsilon \sim \mathcal{N}(0, \mathbf{I})$ is the standard Gaussian noise and $\sigma$ controls the magnitude, we have the first-order Taylor expansion $\mathcal{M}(x + \sigma\epsilon) \approx \mathcal{M}(x) + \nabla_x \mathcal{M}(x)^T \sigma\epsilon$ on perturbed model discrepancy $\mathcal{M}(x + \sigma\epsilon)$, as formulated in Eq.(13), thus we can obtain the expectation of the discrepancy change $\Delta\mathcal{M} = \mathcal{M}(x + \sigma\epsilon) - \mathcal{M}(x)$ below:

$$\mathbb{E}_\epsilon[\Delta\mathcal{M}] \approx \mathbb{E}_\epsilon[\nabla_x \mathcal{M}^T \sigma\epsilon] = \sigma\nabla_x \mathcal{M}^T \cdot \mathbb{E}_\epsilon[\epsilon] = 0, \tag{18}$$

where $\mathbb{E}_\epsilon[\epsilon] = 0$; $\sigma$ and $\nabla_x \mathcal{M}^T$ are constants. Building on that, we can further obtain the variance of $\Delta\mathcal{M}$ below:

$$\begin{aligned}
\mathrm{Var}_\epsilon[\Delta\mathcal{M}] &= \mathbb{E}_\epsilon[(\Delta\mathcal{M})^2] - (\mathbb{E}_\epsilon[\Delta\mathcal{M}])^2 \\
&\approx \mathbb{E}_\epsilon[(\nabla_x \mathcal{M}^\top \sigma\epsilon)^2] \quad (\mathbb{E}_\epsilon[\Delta\mathcal{M}] = 0) \\
&= \sigma^2 \mathbb{E}_\epsilon[(\nabla_x \mathcal{M}^\top \epsilon)^2] \\
&= \sigma^2 \mathbb{E}_\epsilon[\epsilon^\top (\nabla_x \mathcal{M} \nabla_x \mathcal{M}^\top)\epsilon] \\
&= \sigma^2 \mathrm{Tr}(\nabla_x \mathcal{M} \nabla_x \mathcal{M}^\top) \quad (\text{since } \mathbb{E}_\epsilon[\epsilon^\top A\epsilon] = \mathrm{Tr}(A) \text{ for } \epsilon \sim \mathcal{N}(0, \mathbf{I})) \\
&= \sigma^2 ||\nabla_x \mathcal{M}||^2.
\end{aligned} \tag{19}$$

Eq.(19) implies that the variance of the discrepancy change under Gaussian perturbation is proportional to the squared norm of the gradient, *i.e.*, $\mathrm{Var}_\epsilon[\Delta\mathcal{M}] \propto ||\nabla_x \mathcal{M}||^2$, thus the Gaussian noise can automatically induce larger discrepancy changes in regions where the gradient is large, thereby achieving a similar objective to gradient perturbation without computing per-sample gradient.

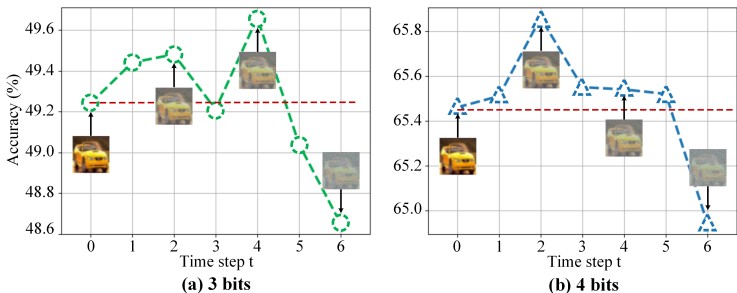

Figure 7: Motivation experiments for classification with varied $Q$: (a) 3 bits and (b) 4 bits on CIFAR-100. The diffused samples within a certain range exhibit better adaptability to $Q$ than the off-the-shelf synthetic samples ($t = 0$) from the existing arts (*i.e.*, IntraQ Zhong et al. (2022)), while the diffused samples from early time steps tend to possess the desirable adaptability to $Q$.

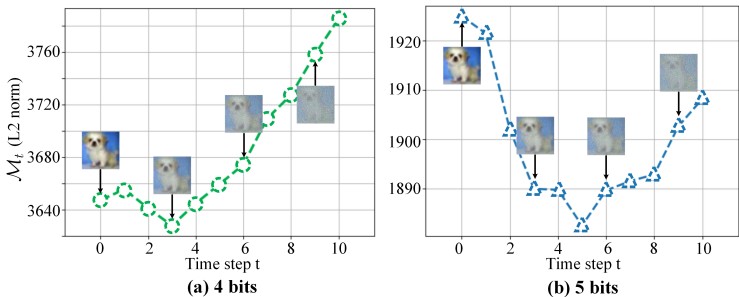

Figure 8: The gap study for varied $Q$: (a) 4 bits, (b) 5 bits. $\mathcal{M}_t$ fluctuates by initially decreasing for early steps and then raising up for late steps of diffusion process, implying that the sample adaptability is rapidly declining starting from the lowest point ($t = 3$ and $t = 5$), owing to the excessive noise. The classification results are from CIFAR-100.

Therefore, the forward diffusion process is considered as an optimal noise strategy for obtaining perturbed samples that minimize $|\mathcal{M}(x_t) - \mathcal{M}^*|$.

$\square$

### A.4    More Discussions on Sample Diffusion Process in Sec.2.3

As discussed in Sec.2.3, on top of the off-the-shelf synthetic samples synthesized via noise optimization strategies, the progressive diffusion process is exploited to obtain a series of desirable diffused samples for varied quantized network ($Q$). Intuitively, the synthetic samples are closely related to the ground-truth sample distribution from the pre-trained full-precision network ($P$), while most of the synthetic samples may be useless for the optimization of $Q$, owing to the large gap between $P$ and $Q$ incurred by the quantization error, implying the poor sample adaptability to $Q$. As aforementioned in Sec.2.3, the diffusion process *progressively* adds noise to synthetic samples to *eliminate* the information, which causes the output difference between $P$ and $Q$, to be *adaptive* to $Q$, as demonstrated in the *early stage* of Fig.7 and Fig.8, *e.g.*, from $t = 1$ to 5 in Fig.7(b) and Fig.8(b); with the time steps raising up, more and more noise are added to the synthetic samples especially the *late stage* of diffusion process, which inevitably reduces the useful information even useless to $Q$, hence becomes *not adaptive* to $Q$, *e.g.*, $t = 4$ to 10 and $t = 6$ to 10 in Fig.8. Therefore, the diffused samples are essentially obtained by properly removing the meaningless information in the synthetic samples for $Q$, thus enhancing the *sample adaptability* to $Q$. *In particular*, high-bit $Q$ (*e.g.*, 4 bits and 5 bits) prefers the diffused samples with a larger time step than low-bit $Q$ (*e.g.*, 2 bits and 3 bits). As discussed in Sec.1, the major reason is that low-bit $Q$ prefers "*easy samples*", *i.e.*, yielding the small output difference between $P$ and $Q$, generally occurred at earlier time steps, owing to its limited representation ability incurred by the quantization error; in contrast, high-bit $Q$ prefers "*hard samples*", *i.e.*, yielding the large output difference between $P$ and $Q$, generally occurred at late time steps. In a nutshell, during the *progressive* diffusion process, less noise may yield "easy samples"

---

**Algorithm 1: DiffDFQ**: Diffusion-Guided Data-Free Quantization

---

**Input** : Initial learning rate $\eta$,
  $P$ and $Q$ parameterized by $\theta_p$ and $\theta_q$
**Output:** Calibrated $Q$

1   **# Sample Synthesis**
2   **for** *epoch* $\leftarrow 1$ **to** *Maximum* **do**
3     Estimating $\mathcal{L}_G$ via Eq.(3)
4     Synthesizing the off-the-shelf synthetic samples $x$ via noise optimization

5   **# Sample Diffusion**
6   **for** $t \leftarrow 1$ **to** $T$ **do**
7     Exploiting $x$ (*i.e.*, $x_0$) to obtain the diffused samples $x_t$ via Eq.(10)
8   Obtaining a series of diffused samples $x_1, x_2, ..., x_T$

9   **# Progressive Network Calibration**
10   Selecting the desirable diffused samples $x_1, x_2, ..., x_{t_d}$ for updated $Q$ and set the initial samples
     $x_t = x_{t_d}$
11   **for** *epoch* $\leftarrow 1$ **to** *Maximum* **do**
12     **if** $t > 0$ *with desirable training epochs allocation strategies* **then**
13       $x_t = x_{t-1}$
14     Estimating $\mathcal{L}_{DiffQ}$ via Eq.(11)
15     Updating $\theta_q \leftarrow \theta_q - \eta \frac{\partial \mathcal{L}_{DiffQ}}{\partial \theta_q}$
16     Updating $\eta$
17   **Return:** $Q$

---

for both low-bit and high-bit $Q$, to enjoy the same output between $P$ and $Q$; while massive noise boosts the "hardness" of the diffused samples, to yield the large output difference between $P$ and $Q$, which is consistent with *our analysis* in Sec.2.3.

In summary, Fig.7 reveals the followings: *the diffused samples within a certain range* $(x_t, x_{t-1}, ..., x_0)$ *of early time steps can better balance reconstruction and adaptability information than* $x$; *the desirable samples should dynamically progress from* $x_t$ *to* $x_0$ *as* $Q$ *is updated, to ensure effective balance between reconstruction and adaptability information, since updating* $Q$ *requires more reconstruction information to improve itself*. In contrast, Fig.8 further verifies that *not all diffused samples are adaptive to* $Q$; in other word, *merely part of all diffused samples near the beginning of the diffusion process are capable of serving as desirable samples to* $Q$; besides, *the selected diffused samples are determined according to the stability of the model discrepancy* $\mathcal{M}$, *since* $\mathcal{M}$ *can remain stable when the diffused samples lie within optimal perturbation range, as indicated in Eq.(7)*.

## A.5   FURTHER DISCUSSION ON DIFFERENT SAMPLE SELECTION STRATEGIES

As mentioned in Sec.2.4, we further discuss the different sample selection strategies for varied $Q$, where **DiffDFQ** prefers adaptive strategy for low-bit $Q$ and non-uniform strategy for high-bit $Q$. Specifically, we perform the experiments with ResNet-18 (ResNet-20) serving as $P$ and $Q$ on ImageNet (CIFAR-10), by applying non-uniform and adaptive strategy to high-bit and low-bit $Q$, as reported in Tab.5. We observe that the adaptive strategy obtains the comparable performance to the non-uniform strategy for varied $Q$, since the adaptive strategy is essentially a special non-uniform strategy.

Notably, for low-bit $Q$, *e.g.*, 3 bits, the adaptive strategy achieves the superior accuracy over the non-uniform one; while for high-bit $Q$, *e.g.*, 4 bits and 5 bits, it is reverse, where either the original synthetic samples ($t = 0$) or part of the diffused samples ($t = 5$, $t = 8$ and $t = 1$) actually fail to be selected for optimizing $Q$ via the adaptive strategy. The *reason* is that, for low-bit $Q$, the adaptive strategy works well at *very* early time steps of diffusion process, *since less noise degraded for smaller time steps easily leads to stable convergence*, as discussed in Sec.2.4. Such fact confirms the *benefits* and *necessity* of combining different sample selection strategies for $Q$ with varied bits as illustrated in Algorithm.1, which is consistent with *our analysis* in the main body.

Table 5: Classification accuracy (%) under different selection strategies. *m*w*n*a indicates the weights and activations are quantized to *m*-bit and *n*-bit. **Selected samples** are indicated by the time step (t) of the diffused samples selected via different sample selection strategies during the calibration process. The best results are reported with **boldface**.

| Dataset | Bit width ($m$w$n$a) | Adaptive strategy | Selected samples | Non-uniform strategy | Selected samples |
|---------|----------------------|-------------------|------------------|----------------------|------------------|
| **CIFAR-10** | $3$w$3$a | **84.74** | $t = 1, 0$ | 84.50 | $t = 1, 0$ |
| | $4$w$4$a | 91.47 | $t = 5$ | **91.97** | $t = 5, 4, ..., 0$ |
| | $5$w$5$a | 93.31 | $t = 8$ | **93.63** | $t = 8, 7, ..., 0$ |
| **ImageNet** | $3$w$3$a | **44.46** | $t = 1, 0$ | 43.72 | $t = 1, 0$ |
| | $4$w$4$a | 66.20 | $t = 1, 0$ | **66.26** | $t = 1, 0$ |
| | $5$w$5$a | 69.85 | $t = 1$ | **69.96** | $t = 1, 0$ |

### A.6 How Do Different Sample Selection Strategies Benefit DiffDFQ?

We also validate varied selection strategies for the diffused samples in Sec.2.4 as random selection, uniform selection, non-uniform selection and adaptive selection with IntraQ+DiffDFQ. Tab.6 reports that the non-uniform strategy (66.12%) upgrades beyond the random (65.72%) and uniform (65.97%) strategy; while maintains a slight disparity with the adaptive strategy (66.18%), implying the importance of different selection strategies for varied $Q$ as remarked in Sec.2.4, which confirms that, *with Q updated, the adaptability of the diffused samples at varied time steps to Q is dynamically changing and the diffused samples from early diffusion process tend to possess better reconstruction information for updated Q* in Sec.2.3.

Table 6: Ablation study for classification accuracy (%) about different selection strategies on CIFAR-100. *n*w*n*a indicates the weights and activations are quantized to *n*-bit. The best results are reported with **boldface**.

| Selection strategy | $3$w$3$a | $4$w$4$a |
|--------------------|----------|----------|
| Random | 49.28 | 65.72 |
| Uniform | 49.48 | 65.97 |
| Non-uniform | 49.64 | 66.12 |
| Adaptive | **50.44** | **66.18** |

### A.7 Additional Results on the CIFAR Datasets

In this section, we provide additional experimental results on CIFAR-10/100, as reported in Tab.7, and summarize our findings below: 1) **DiffDFQ** achieves a significant and consistent accuracy gain over the state-of-the-arts for $Q$ under varied bit-widths, confirming the intuition of rectifying the distribution reconstruction of off-the-shelf synthetic samples with no generator to derive the desirable samples for varied $Q$ via the diffusion process. Specifically, **DiffDFQ** receives at most 7.67% accuracy gains on CIFAR-10/100 over **IntraQ** and **HAST**. 2) **DiffDFQ** delivers the significant improvements for $Q$ with ultra-low bit widths, verifying the *importance* of rectifying reconstruction and adaptability information for $Q$ varied bit widths. Notably, for 3w3a, **DiffDFQ** exhibits the obvious accuracy advantages at most 7.67% and at least 2.19% over **IntraQ**.

Table 7: Classification accuracy (%) comparison with the state-of-the-art over ResNet-20 on CIFAR-10 and CIFAR-100. †: the results reproduced by author-provided code. *n*w*n*a indicates the weights and activations are quantized to *n*-bit and *n*-bit. The best results are reported with **boldface**.

| Dataset | Bit width ($m$w$n$a) | IntraQ Zhong et al. (2022) (CVPR) | IntraQ + DiffDFQ (Ours) | Gains | HAST Li et al. (2023) (CVPR) | HAST + DiffDFQ (Ours) | Gains |
|---------|----------------------|-----------------------------------|-------------------------|-------|------------------------------|------------------------|-------|
| **CIFAR-10** | $3$w$3$a | 77.07 | **84.74±0.09** | +7.67 | 88.34 | **88.52±0.07** | +0.18 |
| | $4$w$4$a | 91.49 | **91.97±0.10** | +0.48 | 92.36 | **92.42±0.03** | +0.06 |
| | $5$w$5$a | 93.51† | **93.63±0.03** | +0.12 | 93.62† | **93.72±0.02** | +0.10 |
| **CIFAR-100** | $3$w$3$a | 48.25 | **50.44±0.45** | +2.19 | 53.98† | **55.08±0.32** | +1.10 |
| | $4$w$4$a | 64.98 | **66.09±0.09** | +1.11 | 66.68 | **66.96±0.03** | +0.28 |
| | $5$w$5$a | 68.72† | **69.05±0.05** | +0.40 | 69.11† | **69.30±0.07** | +0.19 |

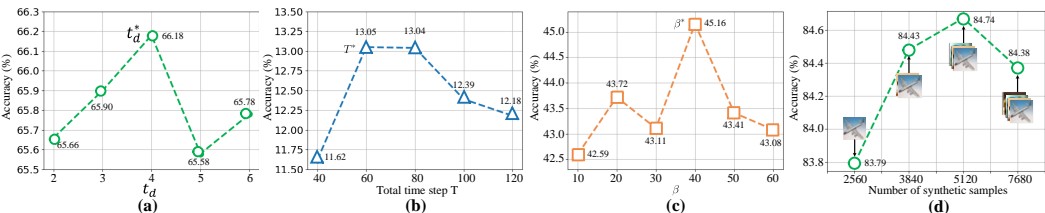

Figure 9: Ablation studies about (a) the selected maximum time step $t_d$, (b) the total time step T, (c) the balance parameter $\beta$, and (d) the amount of synthetic samples.

## A.8  ADDITIONAL PARAMETER STUDIES

As mentioned in Sec.3.1, due to page limitation, we provide more discussions on the key hyper-parameters of our DiffDFQ.

### A.8.1  $t_d$ IN SEC.2.4

$t_d$ serves as the maximum time step of selected diffused samples from sample diffusion process, which is determined by the stability of the model discrepancy $\mathcal{M}$. To verify its effectiveness, we perform the ablation experiments on CIFAR-100 under 4-bit precision by *manually* setting varied $t_d \in \{2, 3, 4, 5, 6\}$, where ResNet-20 serves as $P$ and $Q$. Fig.9(a) illustrates that our **DiffDFQ** obtains the best result given $t_d^* = 4$ as adopted in the main experiments, confirming *advantages* of the automatic $t_d$ over the diffused samples compared to other cases.

### A.8.2  TOTAL TIME STEP T

To validate the effectiveness of T that serves as the number of the total time step for the diffusion process, we verify varied T, *i.e.*, 40, 60, 80 (as adopted in the main experiments), 100 and 120, while perform the experiments with 2-bit precision, where MobileNet-V2 is adopted as $P$ and $Q$ on ImageNet. Fig.9(b) shows that the optimal performance is achieved with $T^* = 60$, implying that carefully adjusting T can earn the extra accuracy gains. Such fact, in turn, confirms the *necessity* of rectifying the distribution reconstruction of synthetic samples to yield desirable samples for varied $Q$ via the forward diffusion process.

### A.8.3  $\beta$ IN EQ.(11)

We further investigate the effect of $\beta$ in Eq.(11), which is utilized to balance the Cross-Entropy loss and Kullback-Leibler divergence loss during the calibration process. We testify varied $\beta \in \{10, 20, 30, 40, 50, 60\}$ to conduct the experiments under 3-bit precision with ResNet-18 serving as $P$ and $Q$ on ImageNet. Fig.9(c) illustrates that the peak appears at $\beta^* = 40$ rather than $\beta = 20$ adopted in the main experiments, *highlighting* the reliance on the knowledge distilled from $P$. The above further *confirms* the effectiveness of the knowledge distillation strategy for optimizing $Q$ over the selected desirable diffused samples for varied $Q$.

### A.8.4  DISCUSSION ON AMOUNT OF SYNTHETIC SAMPLES

To verify the adaptability of *few* off-the-shelf synthetic samples, we study the impact of amount of synthetic samples during the sample synthesis on the sample adaptability, with varied sample numbers as $\{2560, 3840, 5120, 7680\}$. Fig.9(d) illustrates the performance loss as the sample number decreases since less samples fails to well reconstruct the ground-truth sample distribution; while the improvements are not achieved with the sample number increasing. Such fact confirms *the advantages of **DiffDFQ** in rectifying the inherent distribution reconstruction of synthetic samples via sample adaptability*, as discussed in Sec.2.2.

### A.8.5  VARIANCE SCHEDULE $\beta_t$ IN THE FORWARD DIFFUSION PROCESS

In the forward diffusion process, $\beta_t$ controls the speed of noise diffusion, which is crucial to the adaptability of diffused samples to $Q$. We study different noise schedules for $\beta_t$, *i.e.*, linear schedule

and cosine schedule, and perform the experiments with MobileNet-V2 serving as $P$ and $Q$ on ImageNet. Tab.8 indicates that the linear schedule (as adopted in the main experiments) exhibits better performance than the cosine schedule with varied $Q$, since the linear schedule can add the noise *faster* during the diffusion process, so as to earlier yield a series of desirable diffused samples for varied $Q$.

Table 8: Ablation study for classification accuracy (%) about the noise schedule during the forward diffusion process. *n*w*n*a indicates the weights and activations are quantized to *n*-bit and *n*-bit. The best results are reported with **boldface**.

| Noise schedule | *2w2a* | *2w4a* | *3w3a* | *4w4a* | *5w5a* |
|:---:|:---:|:---:|:---:|:---:|:---:|
| **Linear** | **13.04** | **51.95** | **56.15** | **68.62** | 71.61 |
| **Cosine** | 12.20 | 51.13 | 55.35 | 68.58 | **71.65** |

### A.8.6 $D_{gap}$ IN EQ.(5)

$D_{gap}$ in Eq.(5) is exploited to measure the difference between the classification outputs from $P$ and $Q$. To validate the effectiveness of the metric, we further perform the ablation experiments under 4-bit precision by setting varied $D_{gap}$ as KL divergence, $\ell_1$ norm and $\ell_2$ norm (as adopted in the main experiments), where ResNet-18 (ResNet-20) is adopted as $P$ and $Q$ on ImageNet (CIFAR-100); as reported in Tab.9. It is observed that **DiffDFQ** with $\ell_2$ norm outperforms other cases with a large margin (at least 0.11% and 0.19%), implying that $\ell_2$ norm can capture the difference between the classification outputs from $P$ and $Q$ more *sensitively*, given the diffused samples with desirable adaptability to $Q$.

Table 9: Ablation study for classification accuracy (%) about the metric $D_{gap}$ on CIFAR-100 and ImageNet. The best results are reported with **boldface**.

| Metric ($D_{gap}$) | CIFAR-100 | ImageNet |
|:---:|:---:|:---:|
| **KL divergence** | 66.01 | 66.09 |
| $\ell_1$ **norm** | 65.92 | 65.96 |
| $\ell_2$ **norm** | **66.18** | **66.28** |

## B ETHICS STATEMENT

This work adheres to the ICLR Code of Ethics. In this study, no human subjects or animal experimentation was involved. All datasets used, including **CIFAR-10** and **CIFAR-100** Krizhevsky (2009), apart from **ImageNet** (**ILSVRC2012**) Russakovsky et al. (2015), were sourced in compliance with relevant usage guidelines, ensuring no violation of privacy. We have taken care to avoid any biases or discriminatory outcomes in our research process. No personally identifiable information was used, and no experiments were conducted that could raise privacy or security concerns. We are committed to maintaining transparency and integrity throughout the research process.

## C REPRODUCIBILITY STATEMENT

We have made every effort to ensure that the results presented in this paper are reproducible. All code and datasets have been made publicly available in the submitted supplementary material package to facilitate replication and verification. The experimental setup, including training steps, model configurations, and hardware details, is described in detail in the paper. We have also provided a full description of our **DiffDFQ**, to assist others in reproducing our experiments. Additionally, all datasets, including **CIFAR-10** and **CIFAR-100** Krizhevsky (2009), apart from **ImageNet** (**ILSVRC2012**) Russakovsky et al. (2015), are publicly available, ensuring consistent and reproducible evaluation results. We believe these measures will enable other researchers to reproduce our work and further advance the field.

## D   THE USE OF LARGE LANGUAGE MODELS

Large Language Models (LLMs) were used to aid in the writing and polishing of this manuscript. Specifically, we used an LLM to assist in refining the language, improving readability, and ensuring clarity in various sections of the paper. The model helped with tasks such as sentence rephrasing, grammar checking, and enhancing the overall flow of the text.

It is important to note that the LLM was not involved in the ideation, research methodology, or experimental design. All research concepts, ideas, and analyses were developed and conducted by the authors. The contributions of the LLM were solely focused on improving the linguistic quality of the paper, with no involvement in the scientific content or data analysis.

The authors take full responsibility for the content of the manuscript, including any text generated or polished by the LLM. We have ensured that the LLM-generated text adheres to ethical guidelines and does not contribute to plagiarism or scientific misconduct.

