# OpenReview forum: "Revisiting Distribution Reconstruction via Sample Adaptability: Diffusion-Guided Data-Free Quantization"
_ICLR.cc/2026/Conference — Submitted to ICLR 2026_

### Official Review · Reviewer_r6c5 · 2025-10-30

**Soundness:** 2
**Presentation:** 1
**Contribution:** 1
**Rating:** 2
**Confidence:** 4

**Summary:**

This paper aims to improve data-free quantization by generating better synthetic samples. Specifically, the paper proposes adding Gaussian noise to the synthetic samples, following the noise scheduler from diffusion models. DiffDFQ first shows that diffusion-like Gaussian noise is an optimal noise strategy for approximating per-sample gradients from the quantization gap between the full-precision model and the quantized model. Then, the paper empirically validates that different levels of noise are needed for different bitwidths. Based on these observations, the paper proposes using synthetic samples with adaptive Gaussian noise for data-free quantization. Experimental results on CIFAR and ImageNet benchmarks show that the proposed method achieves accuracy gains, especially in low-bit QAT settings.

**Strengths:**

* The paper provides a theoretical basis for adding Gaussian noise to approximate gradients between a full-precision model and its quantized variant.
* The proposed method consistently shows better quantization accuracy, empirically validating it as a superior training strategy.
* The paper offers a detailed background section, which helps readers understand the proposed method.

**Weaknesses:**

* The composition of the paper is awkward.

The paper presents a 3-page introduction section that includes detailed background, motivational experimental results, problem formulation, observations, and even anticipated reader questions about the method. Furthermore, the second section after the introduction is the method section, which is combined with the preliminaries. The experiment section also includes details on quantization and related work. This composition not only results in lengthy sections but also makes it difficult for readers to follow. Therefore, this reviewer suggests separating the sections and reducing the amount of detail in the introduction.

* The theoretical basis is questionable.

While this reviewer acknowledges that providing a theoretical basis is a strength of the paper, the conclusion is not directly aligned with the proposed method. Specifically, the conclusion of Proposition 2 should be that using Gaussian noise is optimal, but the paper instead claims it as a forward diffusion process. Although the forward diffusion process also uses Gaussian noise, the detailed proof in Appendix A.3 concerns only Gaussian noise, not the forward diffusion process, which includes a noise schedule. Moreover, in the proof, the forward diffusion process cannot be applied to Equation 18, since the forward diffusion process represents an interpolation between the sample and the noise, not simply the addition of Gaussian noise (i.e., $E[\Delta M]$ is not zero because x differs). Therefore, the proof demonstrates that the proposition applies only to additive Gaussian noise, not to the forward diffusion process.

* The term adaptability should be further clarified.

The paper states that the reconstructed samples are usually adaptive to specific bitwidths rather than all bitwidths. However, the basis for this claim is the lower accuracy of 3-bit quantization compared to 4-bit quantization. This reviewer believes that the lower accuracy of 3-bit quantization is natural and primarily caused by harsher quantization errors, not by sample quality. Moreover, the explanation of the adaptive strategy is too shallow, making the method difficult to reproduce.

**Questions:**

Please refer to the weakness section.
Additionally, here are minor questions and suggestions.
* The citation format of the paper could be improved. This reviewer suggests using parentheses in citations.
* Figure 2 (b): Aadaptability -> Adaptability
* How do easy samples, which induce a small gap between the full-precision model and its quantized variant, achieve better accuracy?
* There are no complete experimental results on the CIFAR dataset, while the ablation study and visual analysis are based on CIFAR. This reviewer believes that including ablation and analysis results on the ImageNet dataset could strengthen the paper, as the main experimental results are derived from ImageNet.

---

### Official Review · Reviewer_JsvL · 2025-10-31

**Soundness:** 2
**Presentation:** 2
**Contribution:** 2
**Rating:** 4
**Confidence:** 5

**Summary:**

In data-free quantization, which is a research field that quantizes target neural networks without real data, synthetic dataset are generated to replace the original dataset and preserve the performance of target models.
Generally, a synthetic dataset is optimized for the target bit setting.
The paper contends that synthesized samples are adaptive to the target bit setting only, and using them to quantize a model to a different bit setting leads to performance degradation.

To resolve this problem, the paper adopts a diffusion forward process that is used to train diffusion models that synthesize realistic images or videos.
The paper proposes to construct additional synthetic datasets by adding noise perturbation to the original synthetic dataset for quantization.
Also, the paper proposes progressive sample selection that uses fewer noise samples step by step while quantization.

By integrating the proposal with prior works, the paper achieves performance improvement on various models and datasets.

**Strengths:**

- The paper defines the problem situation of quantization well that people have not much thought of before.

- It is an original approach that using diffusion process to quantization.

- Compared to various existing studies, the paper shows the performance improvement effect of the proposed method.

**Weaknesses:**

- Even though readers can infer the meanings, the authors should have defined the key terms, such as sample adaptability and distribution reconstruction, with their own words.

- The connection between propositions 1 and 2 is weak. In Proposition 1, the direction of the perturbation should be the same as that of M's gradient. However, perturbations in the diffusion process usually follow a Gaussian distribution. It is difficult to understand the relationship between the argument of proposition 2 and the direction of perturbation by the given content alone.

- The paper executed experiments of the proposed method based on other works. To show the effectiveness of the proposal clearly, it would be better to execute ablation studies without prior works such as IntraQ, HAST, and Genie.

- The paper tackles that the synthetic dataset is adaptive to quantization bit, thus using this dataset to quantize models with other bit settings can harm their performance. However, it seems that there isn't any intervention of a quantization model or quantization bit settings while generating a synthetic dataset generation based on reconstruction.
It runs counter to the argument.

- Typo: In the section 3.3.1, there are consecutive commas like ', ,'.

**Questions:**

- What if a target model is quantized with the whole dataset, including diffused samples of every step?

- According to the paper's argument, the performance of a quantized model to 4 bit would be degraded if it is quantized with a synthetic dataset generated under 3 bit setting.
To strengthen the paper's argument of the problem, can the authors execute other experiments about quantizing the model to 4 bit with 3 bit dataset, and compare it with the opposite setting?

- In the experiments of Section 3.5, the paper contends that the quantization complexity of HAST + DiffDFQ is lower than that of HAST itself.
Does it mean that the experimental settings for them are different? Or does HAST + DiffDFQ stop early before the experiment ends?
According to the experimental setting, the number of epochs is 400, and in this case, the complexity of both methods is the same.
If the intention is that HAST + DiffDFQ converges faster than HAST itself, the reviewer think that it is not a complexity problem.

---

### Official Review · Reviewer_DYmb · 2025-11-03

**Soundness:** 2
**Presentation:** 2
**Contribution:** 3
**Rating:** 4
**Confidence:** 4

**Summary:**

This paper proposes a new data-free quantization (DFQ) method called DiffDFQ. The core idea of this paper is to solve two problems in existing methods: (1) the data synthesized by the existing methods is only applicable to a special bit width. Once it is used for the ultra-low bit width (such as 3-bit), the performance will decline sharply; (2) some methods that try to solve the adaptability problem will damage the quality of synthetic data in order to meet the gap between FP model P and quantized model Q. This paper points out that P and Q should not pursue "zero gap". On the contrary, there is an ideal gap M*. Then, this paper proves that the forward diffusion process (gradually adding noise) is the optimal noise strategy approaching M*. An three-stage algorithm is proposed: (1) sample synthesis: it first synthesize a batch of well reconstructed samples x_0; (2) sample diffusion: perform forward diffusion (plus noise) on x_0 to obtain a "calibration set", which contains a series of samples from x_t (noisy) to x_0 (clean); (3) progressive calibration: wen training Q, it first use the sample with high noise (such as x_t) for training, and then gradually transition to the one with low noise (such as x_0).

**Strengths:**

1. Comprehensive experimental results.

2. The theoretical proof is logical.

**Weaknesses:**

1. The paper proves that diffusion is an efficient disturbance strategy, but it does not (and cannot) prove that the final approximation of this disturbance is the theoretical M*.

2. This method seems specified to solve the 3-bit case. On 4-bit and 5-bit, the benefits of this complex diffusion and progressive calibration can be almost ignored by experimental errors. This has reduced the universality claimed in the paper.

3. Many concepts in this paper make me very confused. I have done my best to read this paper, but I still can not find out the meaning of reconstructed samples and distribution reconstruction. Also, what do you mean when you say ‘AdaDFQ achieves the enhanced adaptability across varied bit widths, but compromises the fidelity of original reconstructed information for Q∗’? How do you support this argument?

4. This paper seems to generate x_0 at first and then add noise to it. Is that so? This paper also claims that this method shows a generalization across bit-widths. How did you embody this configuration in your experiment?

I do believe this paper need to be polished greatly to make it easy to read. The current version is not suitable for the reader following. I tend to reject this paper, not only for it vague writing but also for some technical flaws.

**Questions:**

Please see the weakness part.

---

### Official Review · Reviewer_2WQw · 2025-11-03

**Soundness:** 2
**Presentation:** 2
**Contribution:** 2
**Rating:** 4
**Confidence:** 3

**Summary:**

This paper addresses a key challenge in Data-Free Quantization (DFQ) , where synthetic data used to calibrate a quantized network (Q) often fails to generalize across different bit widths, particularly for ultra-low bit (e.g., 2 or 3-bit) models. The authors observe that existing synthetic samples lack "sample adaptability," and methods that optimize for adaptability tend to compromise distribution reconstruction fidelity. They propose Diffusion-Guided Data-Free Quantization (DiffDFQ), a three-stage approach. First, it uses an existing method to synthesize reconstructed samples. Second, it applies a forward diffusion process (progressively adding noise) to these samples to create a "calibration set" of diffused samples at various noise levels. Third, it calibrates the quantized network using a "progressive sample selection strategy" that utilizes this diffused sample set. The authors claim this diffusion process rectifies the samples, striking an effective balance between reconstruction and adaptability, thus yielding "desirable samples" that improve performance for Q, especially at ultra-low bit widths.

**Strengths:**

1. This paper provide significant performance improvement on 3-bits.
2. The paper includes several useful ablation studies that validate the contributions of its core components

**Weaknesses:**

1. The writting of this paper is poor. It takes me long time to understand the writting even I am a expert in quantzization. The writing should be re-organized.
2. This paper show negligible performance improvement on 4-bits, but only achieve benefit on 3-bit. However, I doubt the practical usage of 3-bit data free quantization in real scnarios. In my mind, 3-bit quantization suffers from significant performance degeneration and can not be directly use. Therefore, 3-bit model has to rely on QAT on real samples.
3. The claim of improved training efficiency is based on a single cherry-picked data point, while other results show the benefit is marginal and not universal. The paper boasts that DiffDFQ reduces convergence time by "at most 35.5%" , citing the 4-bit case (18.96h vs 12.23h). However, the same table shows that for the 5-bit case, the time-saving is a mere 11.2% (16.83h vs 14.94h). Similar to the accuracy gains, this suggests the claimed efficiency benefit is not general but is instead another best-case result from a specific configuration, while the gains diminish at other bit-widths.
4. The "Progressive Calibration" stage is just standard curriculum learning, but the paper fails to acknowledge this or position its work within that field.

**Questions:**

Please refer weakness for detials.

---

### Meta-Review · Area_Chair_4YGv · 2025-12-06

**Summary:**

The paper explores diffusion-based data-free quantization, but all reviewers raise substantial concerns. The writing is unclear, key concepts are not well defined, and the theoretical claims do not convincingly justify the method. Empirically, meaningful gains appear only at 3-bit, with negligible improvement at 4/5-bit, limiting practical relevance. Several core claims (adaptability, efficiency, optimality of diffusion) are not sufficiently supported by experiments. Given the unclear exposition, weak theoretical grounding, and limited generality of results, I recommend rejection.

**Reviewer Concerns:**

The authors did not submit a rebuttal. Therefore, none of the reviewers’ concerns have been addressed, and all original issues raised in the reviews remain outstanding.

**Reviewer Scores:**

No rebuttal was submitted and no further discussion occurred. Therefore, I believe none of the reviewers would have changed their scores.

---

### Decision · Program_Chairs · 2026-01-26

Reject